# Proximal tubular renal dysfunction among HIV infected patients on Tenofovir versus Tenofovir sparing regimen in western Kenya

**Mercy Jelagat Karoney**🆔*, **Mathew Kirtptonui Koech, Evangeline Wawira Njiru, Willis Dixon Owino Ong'or**

Department of Medicine, College of Health Sciences, School of Medicine, Moi University, Eldoret, Kenya

* karoneymercy@gmail.com

## Abstract

### Introduction

Tenofovir Disoproxil Fumarate (TDF) is the most widely used Anti-Retroviral Therapy (ART) drug due to its potency, safety profile and World Health Organization (WHO) recommendation. TDF causes proximal tubular renal dysfunction (PTRD) leading to Fanconi syndrome, acute kidney injury and chronic kidney disease. Modest rates (2–4%) of TDF related toxicity based on estimated Glomerular Filtration Rate (GFR) have been described, while TDF-induced PTRD has been reported to be 22%. TDF toxicity is more likely among African patients, it is reversible and TDF may be renal dosed in patients with dysfunction. The objective of this study was to assess proximal tubular renal dysfunction, global renal function, and their determinants among patients on TDF versus TDF-sparing regimen.

### Methods

This was a cross-sectional study among people living with HIV/AIDS (PLWHA) attending the Academic Model Providing Access to Healthcare (AMPATH) program. The primary outcome of interest in this study was PTRD while the secondary outcome of interest was estimated GFR. PTRD was defined as any two of beta-2 microglobulin in urine, metabolic acidosis, normoglycemic glucosuria and fractional excretion of phosphate. Student's t-test, chi-square and their non-parametric equivalents were used to test for statistical significance. Univariate and multivariate logistic regression analysis was carried out.

### Results

A total of 516 participants were included in the final analysis, 261 on TDF while 255 were on TDF-sparing regimens. The mean (SD) age of all participants was 41.5 (12.6) years with majority being female (60.3%). The proportion of PTRD was 10.0% versus 3.1% in the TDF compared to TDF-sparing group (P<0.001). Mean estimated GFR was 112.8 (21.5) vs 109.7 (21.9) ml/min/1.73mm$^3$ (P = 0.20) for the TDF compared to TDF-sparing group. TDF users were more likely to have PTRD compared to non-TDF users, adjusted Odds Ratio (AOR) 3.0, 95% CI 1.12 to 7.75.

**Data Availability Statement:** All relevant data are within the paper and its Supporting Information files.

**Funding:** MJK had a career development grant from EDCTP (European and Developing Countries Clinical Trials Partnership) https://www.edctp.org/ Grant number: TMA2015CDF-1002 The funders had no role in study design, data collection and analysis, decision to publish, or preparation of the manuscript.

**Competing interests:** The authors have declared that no competing interests exist.

**Abbreviations:** AMPATH, Academic Model Providing Access to Healthcare; ART, Antiretroviral Therapy; B2M, Beta-2 microglobulin; BMI, Body Metabolic Index; CD4, Cluster of Differentiation 4; CG, Cockcroft- Gault; CKD-EPI, Chronic Kidney Disease Epidemiology Collaboration; EDCTP, European and Developing Countries Clinical Trials Partnership; FDA, Food and Drug Administration; FEphos, Fractional Excretion of Phosphate; GFR, Glomerular Filtration Rate; HIV, Human Immunodeficiency Virus; KDIGO, Kidney Disease: Improving Global Outcomes; MTRH, Moi Teaching and Referral Hospital; PLWHA, People Living with HIV/AIDS; PTRD, Proximal Tubular Renal Dysfunction; TDF, Tenofovir Disoproxil Fumarate; WHO, World Health Organization.

## Conclusion

There was significant PTRD in the TDF compared to TDF-sparing group without significant difference in estimated GFR. The clinical significance of these findings may not be clear in the short term.

## Introduction

Renal disease associated with HIV infection has multifactorial causes including HIV itself, co-infections, co-morbidities and their treatment [1]. Antiretroviral therapy (ART) use has led to improvements in HIV and renal related outcomes [2]. Some ART drugs have however been noted to cause renal toxicity through tubular and interstitial damage, and through drug interactions with other concomitant medications [3, 4]. Since the advent of ART, HIV patients are living longer thus non-infectious co-morbidities and renal toxicities have become important areas of research and contributors to morbidity [5]. Tenofovir Disoproxil Fumarate (TDF) was recommended by World Health Organization (WHO) 2013 guidelines as the first line of therapy in combination with other anti-retroviral drugs [6].

TDF causes Fanconi syndrome, acute kidney injury or chronic kidney disease by through proximal tubular injury [7, 8]. Modest rates of TDF-related renal dysfunction have been described in literature with 1–2% of renal dysfunction reported [9, 10]. Most studies however report global kidney function using estimated glomerular filtration rate (GFR) yet early detection of TDF-associated nephrotoxicity requires testing for proximal tubular renal dysfunction (PTRD) [11]. Studies investigating proximal tubular dysfunction report a high prevalence of subclinical dysfunction, ranging from 15–22%, in HIV infected patients [12, 13].

Detection of TDF-associated toxicity while it is still early or mild requires specific investigations for proximal tubular injury [11]. Proximal tubular injury can be determined through urinalysis for glucose and protein, serum phosphate and bone fracture rate [14, 15]. Beta-2 microglobulin (B2M) in urine is a sensitive marker for assessing proximal tubular proteinuria [14, 16, 17]. WHO guidelines do not emphasize the need or frequency of monitoring renal function in patients on TDF, leaving this to the discretion of the clinicians. Furthermore, subclinical toxicity is missed when serum creatinine is used to assess the global renal function [6]. The objective of this study was to assess proximal tubular renal dysfunction and mean GFR among HIV-infected patients on TDF regimen compared to those on TDF-sparing regimen and the factors associated with PTRD.

## Methods

### Study design

This was a cross-sectional design comparing outcomes (proximal tubular renal dysfunction and global renal function) in TDF (exposed) and TDF-sparing (unexposed) groups. The study was carried out between 1st September 2016 and 30th September 2019.

### Study setting

The study was carried out at the ambulatory HIV care clinic at Moi Teaching and Referral Hospital (MTRH) as provided by the Academic Model Providing Access to Healthcare (AMPATH) program. AMPATH program is collaboration between Moi Teaching and Referral Hospital, Moi University College of Health Sciences, and a group of North American

academic medical centers led by Indiana University. The program has enrolled over 160,000 HIV-positive patients in over 144 clinical sites in both urban and rural western Kenya over the last 15 years.

## Study participants

The target population comprised of HIV-infected persons attending AMPATH's MTRH clinics in western Kenya. The results of this study are generalizable to all HIV-infected patients within the MTRH catchment area in western Kenya. Approximately 12,000 HIV infected patients on ART are enrolled in AMPATH's urban MTRH clinic with about 3,000 seen monthly.

Participants for this study were selected through stratified random sampling. Participants identified from the sampling technique above were checked for eligibility. Participants who had an abnormal baseline creatinine at initiation of ART and those with known renal disease were excluded.

## Variables

**Dependent variables.**   Dependent variables for this study were PTRD and estimated GFR which represented the overall renal function. GFR was calculated from serum creatinine and age of the participants by CKD-EPI formula on a Microsoft Excel spreadsheet before being merged with the other variables.

**Independent variables.**   Socio-demographic and clinical variables collected were age, gender, co-morbidities, concomitant use of nephrotoxic medication and body mass index (BMI). HIV disease status included information such as duration of ART use, WHO clinical staging, most recent viral load and CD4 count at baseline.

Confounders determined *a priori* for this study were age, gender, co-morbidities, duration of ART, HIV status and concomitant medications. Data was collected on drugs known to cause proximal tubular toxicity including aminoglycoside antibiotics, antifungal agents such as amphotericin B and anticancer drugs such as cisplatin [18].

## Data sources and measurement

Participants' socio-demographic characteristics and disease status were collected by questionnaire and data collection sheet. Blood pressure, height and weight and blood sugar were measured before participants were taken to the lab. Blood and urine specimens were collected from each participant for the lab tests needed; beta-2 microglobulin in urine, urinary creatinine, urinary phosphate, urinary glucose and serum creatinine, serum phosphate and serum glucose. All the phlebotomy procedures were carried out under sterile conditions.

**Proximal tubular renal dysfunction** was defined as any 2 out of 4 parameters including normoglycemic glucosuria, metabolic acidosis, beta-2 microglobulinuria, and fractional excretion of phosphate >20%.

**Normoglycemic glucosuria** in this study was defined by detectable glucose in urine by dipstick despite a random blood glucose of less than 11.1mmol/l.

**Metabolic acidosis** was defined as plasma bicarbonate less than 20mmol/l.

**Tubular proteinuria** was defined as presence of excessive amounts of beta-2 microglobulin in urine more than 0.3mg/mmol.

**Phosphate wasting:** Phosphate wasting was defined as a fractional excretion of phosphate (FEphos) of >20% among participants if normal serum phosphate levels (0.85 to 1.45 mmol/l) or >10% among participants with hypophosphatemia (serum phosphates of <0.85 mmol/l)

[19, 20].

$$FEphos = \frac{\text{urinary phosphate (Up)x plasma creatinine (Pcr)}}{\text{plasma phosphate (Pp)x urinary creatinine (Ucr)}} \times 100$$

**Renal function/ Estimated GFR:** The National Kidney Foundation's Practice Guidelines for Chronic Kidney Disease was used to establish a cut point, eGFR <90 mL/min/1.73 m$^2$, for decreased kidney function [21]. The CKD-EPI creatinine equation is expressed for specified age, sex and serum creatinine level. The equation is GFR = The CKD-EPI equation, expressed as a single equation, is GFR = $141 \times \min (\text{Scr}/\kappa, 1)^{\alpha} \times \max(\text{Scr}/\kappa, 1)^{-1.209} \times 0.993^{\text{Age}} \times 1.018$ [if female] _ 1.159 [if black] [22].

## Study size

A formula for logistic regression was used to determine the minimum sample size required. The (N) based on logistic regression model was obtained using the formula suggested by Peduzzi et al, **N = 10k/p**, where k is the number of independent variables and p is the number or events or prevalence of the condition of interest as determined from previous studies [23]. The number of independent variables in this study were 7 (age, sex, co-morbidities, body weight, concomitant medication, viral load and duration of ARV use). The prevalence was obtained from a study done in Spain that determined the prevalence of proximal tubular dysfunction among infected patients as 15% [13]. Using the Peduzzi formula the sample size required was 467 total participants. Assuming a non-response rate of 10%, the N was inflated by the formula n ÷ (1- non-response rate). The estimated final sample size needed therefore was **518**, 259 exposed and 259 unexposed participants.

## Statistical analysis

Proportions were calculated for PTRD in TDF, and TDF-sparing group then compared using chi-square for statistical significance. Mean and corresponding standard deviations were calculated for the estimated GFR and then Student's t-test was used to compare for statistical significance. Wilcoxon rank sum test was used for non-normally distributed continuous variables while Fisher's exact test was used where frequencies were small. Multiple logistic regression analysis was carried out to determine the factors affecting the association between TDF exposure and PTRD. *A priori* determined confounders: age, sex, co-morbidities were included in the final model regardless of their association with TDF exposure and PTRD.

## Ethical consideration and permission

Ethical approval was obtained from the Institutional Research and Ethics Committee (IREC) of Moi University and MTRH and AMPATH administration. Informed consent was obtained from each participant enrolled into study. Participants were free to withdraw from the study, there were no monetary incentives provided to participate. Results of the participants were communicated back to the primary clinician for necessary action.

# Results

## Recruitment of participants and missing data

A total of 539 participants were approached for recruitment, 529 met the inclusion criteria while 10 were excluded because 3 refused to consent, 6 had known diabetes or overt hypertension and 1 participant was on a second-line regimen (Fig 1). Out of 516 included in the final analysis, 261 were TDF users while 255 were in the non-TDF users.

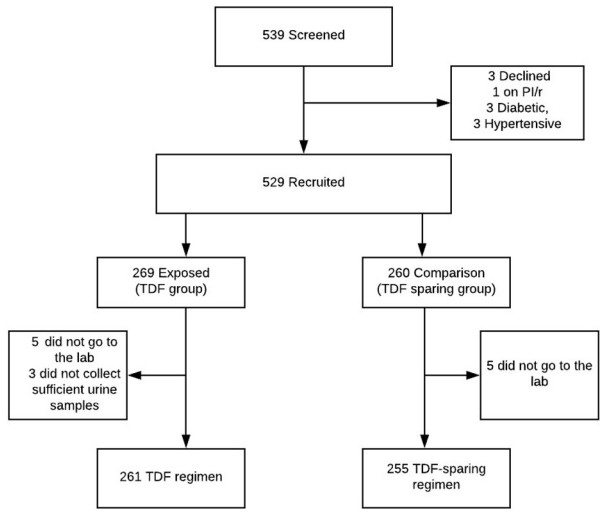

**Fig 1. Recruitment schema.**

Missing data was excluded from the multivariable analysis, and this was not expected to introduce any bias to the analysis because it was a small percentage missing 13/529 (2.5%).

## Socio-demographic and clinical characteristics of the participants

The mean age (SD) of all the 516 participants was 41.5 (12.6) years, with participants ages ranging from age 18 to 79 years. TDF regimen users were younger compared to TDF sparing regimen users with mean age (SD) 39.2 (12.6) vs 43.9 (12.2) (p <0.001). Overall female to male ratio was 3:2, with female 310/516 (60.3%). Majority of the participants had undetectable viral load 423/516 (82.0%). Participants in the TDF-sparing group had more preexisting hypertension and diabetes compared to the TDF regimen group, 13.7% versus 6.5% (p = 0.02). TDF regimen users has used ART for a shorter duration (4.6 years vs 8.0 years p<0.01). Regarding the HIV status, TDF-sparing regimen users had lower baseline CD4 counts (323 vs 370 cells/mm3 p = 0.05), and majority were WHO stage 3 (40% vs 34.5% p = 0.02) compared to TDF-sparing group. There were no statistically significant differences between the groups on use of concomitant potentially nephrotoxic medication, basal metabolic index, and viral load suppression. Table 1 shows the overall sociodemographic and clinical characteristics, comparison by TDF use and p values.

## Proximal renal tubular dysfunction and global renal function

The proportion of participants with PTRD was 26/261 (10.0%) for the TDF regimen group compared to 8/255 (3.1%) for the TDF-sparing group. PTRD was significantly higher in TDF vs TDF-sparing group, Unadjusted Odds ratio 3.42 (95%CI 1.50 to 7.76). The parameters used to determine PTRD are shown in Fig 1 below. TDF users had higher percentage of Metabolic acidosis (41.8 vs 35.7%) and tubular proteinuria (18.8% vs 6.3%) compared to TDF-sparing regimen (Fig 2). Very few participants had nondiabetic glucosuria.

Global renal function was determined by serum creatinine and glomerular filtration rate. The mean estimated GFR (SD) was 112.8 (21.5) vs 109.7 (21.9) ml/min/1.73m$^2$ for the TDF and TDF-sparing group respectively with UOR 1.00 (95% CI 0.99 to 1.01). Although 55/516 (10.7%) of the participants had elevated serum creatinine, this was not significantly different in the two groups.

**Table 1. Sociodemographic and clinical characteristics of adult PLHWA on TDF versus TDF-sparing regimens in Western Kenya, January 2017 to Dec 2019.**

| Participant characteristics | | Total | TDF use | TDF sparing regimen | P value[a] |
|---|---|---|---|---|---|
| | | N = 516 (% or SD) | n = 261 (%) | n = 255 (%) | |
| Age | Mean (SD) years | 41.5 (12.6) | 39.2 (12.6) | 43.9 (12.1) | 0.001* |
| Gender | Male | 206 (39.7%) | 96 (36.4%) | 110 (43.1%)) | 0.115# |
| | Female | 310 (60.3%) | 165 (63.6%) | 145 (56.9%) | |
| Comorbidities | None | 464 (89.9%) | 244 (93.5%) | 220 (86.3%) | 0.02# |
| | Hypertension/Diabetes | 52 (9.6%) | 17 (6.5%) | 35 (13.7%) | |
| Concomitant medication | None | 413 (80.0%) | 207 (79.3%) | 206 (80.8%) | 0.675# |
| | Nephrotoxic | 103 (20.0%) | 54 (20.7%) | 49 (19.2%) | |
| BMI in kg/m$^2$ | Mean (SD) | 23.0 (4.5) | 22.9 (4.7) | 23.1 (4.4) | 0.579# |
| Duration of ART use | Mean (SD) years | 6.3 (3.5) | 4.6 (3.4) | 8.0 (2.7) | 0.001* |
| HIV-1 viral load | Undetectable | 423 (82.0%) | 220 (84.3%) | 203 (79.6%) | 0.166# |
| | Detectable | 93 (18.0%) | 41 (15.7%) | 52 (20.4%) | |
| CD4 at baseline | Mean (SD) cells/mm$^3$ | 346.4 (238) | 370.0 (251) | 323.6 (223) | 0.05* |
| WHO clinical stage | Stage 1 | 184 (35.7%) | 103 (39.5%) | 81 (31.8%) | 0.02# |
| | Stage 2 | 98 (19.0%) | 54 (20.7%) | 44 (17.3%) | |
| | Stage 3 | 192 (37.2%) | 90 (34.5%) | 102 (40.0%) | |
| | Stage 4 | 42 (8.1%) | 14 (5.3%) | 28 (11.0%) | |

Tests used to calculate significance

*Student t test

#Chi-square

[a] P-value ≤ 0.05 is significant

Abbreviations TDF–Tenofovir disoproxil fumarate, SD–standard deviation, BMI–Basal metabolic index, HIV- Human Immunodeficiency virus, CD4- Cluster of differentiation 4, WHO–World Health Organization, ART–Antiretroviral therapy PLWHA- People living with HIV/AIDS

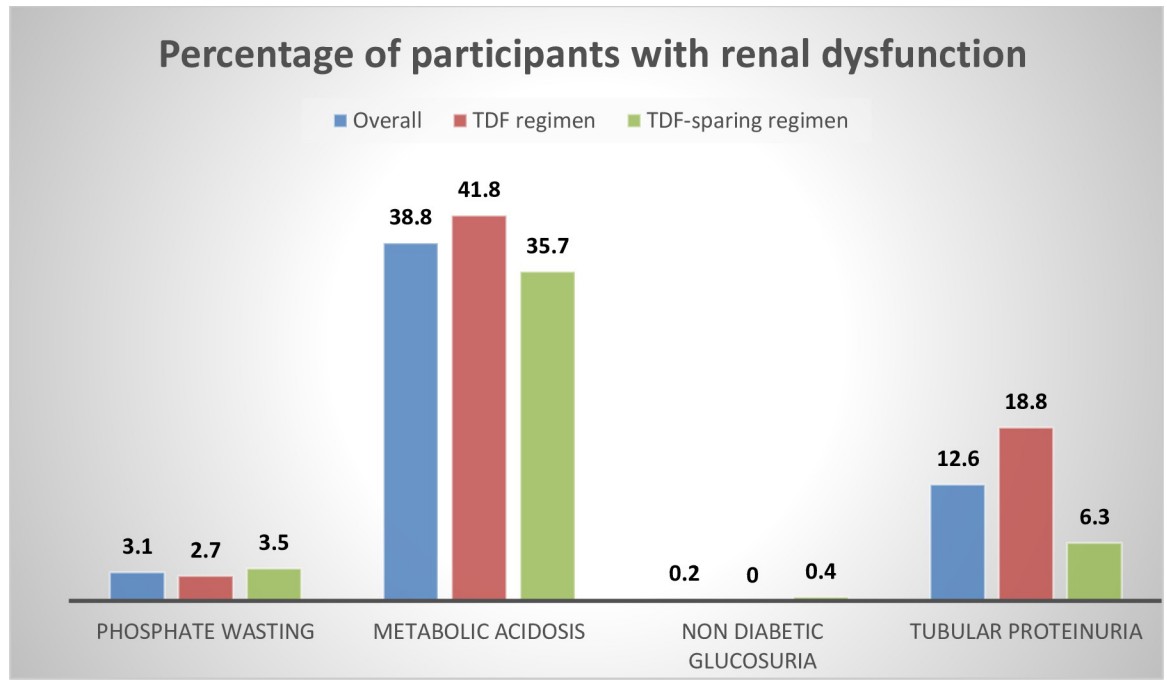

**Fig 2. Percentage of participants with abnormal renal parameters for the overall group as well as categorized by TDF use.**

UOR 1.00 (95%CI 0.99 to 1.01). The mean estimated GFR was found to be 93.3 vs 112.5ml/min/1.73m$^2$ (p = 0.001) for participants who had PTRD vs those who did not have PTRD.

### Factors associated with PTRD

Table 2 below presents the results for the univariate (UOR) and multivariate/adjusted (AOR) logistic regression analysis. TDF-regimen users were 3.41 times more likely to have PTRD compared to TDF-sparing group UOR 3.41 (95%CI 1.52 to 7.69). This relationship between TDF use and PTRD remained positive after adjustment of other factors in the multivariate analysis, Adjusted OR, (AOR) 3.39 (95% CI 1.33 to 8.62). A one-year increase in age was also associated with PTRD in the multivariate analysis with AOR 1.03 (95% CI 1.01 to 1.06).

Female gender, co-morbidities, concomitant use of nephrotoxic medication, increase in BMI, duration of ART and detectable viral load were not associated with increase the likelihood of PTRD in both univariate and multivariate analysis. Table 2 below presents the results for the univariate (UOR) and multivariate/adjusted (AOR) logistic regression analysis.

## Discussion

TDF use in first line regimens has increased since the release of WHO 2013 guidelines. The release of the 2015 guidelines further recommended test and treat strategy for all HIV infected persons. Several studies have demonstrated an increased prevalence of renal tubular dysfunction in TDF-treated patients in comparison with patients receiving other ART regimens. The present study reports a significantly higher proportion (10% vs 3%) of renal tubular dysfunction among use of TDF versus a TDF-sparing regimen. Higher proportion of tubular dysfunction have also been reported in Ghana (35% vs 6%), Germany (17% vs 3%), Spain (22% vs 6%) and France (31% vs 15%) [13, 24–26]. These studies however found much higher prevalence among those on TDF compared to the present study probably because the variations in the populations of study. Such variations may include concomitant use of second line regimens and race such as in the Spanish, French and German cohort. These studies done in Europe studied Caucasian and did not exclude use of second line regimens which are known to worsen TDF toxicity [13, 25, 26]. Further differences in the studies can be explained by immigration of populations at high risk to developed countries. The distribution of apolipoprotein 1 (*APOL1*) risk alleles are highest among individuals from West Africa, intermediate among

**Table 2. Factors associated with PTRD.**

| Participant characteristics | | Unadjusted OR (95%CI) | Adjusted OR (95%CI) |
|---|---|---|---|
| TDF use | No | 1 | 1 |
| | Yes | 3.41 (1.52 to 7.69) | 3.39 (1.33 to 8.62) |
| Age | Years | 1.02 (0.99 to 1.05) | 1.03 (1.01 to 1.06) |
| Gender | Male | 1 | 1 |
| | Female | 0.73 (0.36 to 1.47) | 0.79 (0.37 to 1.69) |
| Co-morbidities | None | 1 | 1 |
| | Yes | 0.67 (0.25 to 1.77) | 0.63 (0.14 to 2.91) |
| BMI | (kg/m$^2$) | 0.99 (0.91 to 1.07) | 0.99 (0.92 to 1.08) |
| Concomitant medication | None | 1 | 1 |
| | Nephrotoxic | 0.67 (0.25 to 1.77) | 0.72 (0.26 to 1.97) |
| Viral load | Undetectable | 1 | 1 |
| | Detectable | 1.42 (0.62 to 3.24) | 1.58(0.66 to 3.79) |
| Duration of ART | Years | 0.89 (0.81 to 1.00) | 0.95 (0.85 to 1.07) |

those from Southern Africa, and lowest among those from East Africa [27]. Therefore, West Africans have higher rates of kidney disease compared to East Africans 20% vs 14% [28].

The non-significant difference in estimated GFR found in this study (112.8 vs 109.7 mL/min/1.73 m$^2$ p = 0.8) has also been demonstrated in Ghana (99 vs 96 mL/min/1.73 m$^2$ p = 0.21), Spain (109 vs 119 mL/min/1.73 m$^2$ p = 0.1), Germany (106 vs 104 p = 0.375) and Canada (104.9 vs 103.5mL/min/1.73 m$^2$ p>0.05). A cohort of Taiwanese HIV-infected patients also demonstrated non-significant annual decline in estimated GFR between persons on TDF and TDF-sparing 2.7 vs 1.8 mL/min/1.73 m$^2$ p = 0.567.

Guidelines on TDF use have been based on several studies that showed no significant renal dysfunction among TDF users compared to TDF-sparing regimens such as the present study. These non-significant results may be the result of short-term duration of TDF use among the participants in this study. TDF toxicity may be remain subclinical for several years before global function as measured by estimated GFR is impaired. Evidence of tubular dysfunction in the absence of change in estimated GFR has been demonstrated in this study.

Statistically significant differences in mean estimated GFR (102 vs 105 mL/min/1.73 m$^2$ p = 0.01) were described in a Ugandan study. The difference in the Ugandan study was however small and may not be clinically significant in making decisions. This difference may have resulted from the use of different way of estimating GFR by the Ugandan study where the authors used Cockcroft-Gault formula. Although there was a statistically significant lower mean estimated GFR for those who had PTRD vs those who did not (93.3 vs 112.5ml/min/1.73m$^2$ p<0.001), the level of GFR was not clinically significant. According to the National Kidney Foundation, this GFR falls in the mild loss to normal range [29].

Previous studies from sub-Saharan Africa identify lower CD4 cell counts, older age and gender as risk factors for significant renal impairment [30–32]. A similar cohort of HIV-infected persons in Tanzania found predictors of renal dysfunction in multivariate analysis include female, BMI, CD4 cell count <200 cells/mm$^3$ and WHO clinical stage II or above [33]. This contrasts greatly with this study which only described normal BMI as a significant related protective factor.

TDF use was significantly associated with increased likelihood of tubular toxicity in this study. This was comparable to studies in Ghana, Spain and Zambia [13, 24]. This was an expected finding because in this study TDF caused tubular injury as previously described in literature.

The major strength of this study was a large sample size that allowed sufficient power for all objectives. a major limitation was the use of non-fasting serum phosphate and spot urinary phosphate levels may have led to the underestimation of the participants with tubular dysfunction in this study.

## Conclusion

In conclusion, there was significant proximal tubulopathy in HIV patients on TDF compared to TDF-sparing regimen. There was no significant difference in the mean estimated GFR in the 2 groups. The median duration of ART use was 6 years in the two groups therefore these findings could vary over longer duration of time. The factors associated with PTRD were TDF use and normal BMI which was found to be protective. Other factors such as age, sex, duration of ART use, viral load and presence of comorbidities were not significantly associated with PTRD despite being selected *a priori* as risk factors. Periodic screening of tubular function parameters should be recommended to patients receiving TDF. A subsequent study to establish the clinical significance of tubular dysfunction in terms of progression to chronic kidney disease and bone loss should be carried out.

## Supporting information

**S1 Dataset.**
(XLS)

## Acknowledgments

We acknowledge all the patients and TREND research team, AMPATH reference lab team and the Lancet lab.

## Author Contributions

**Conceptualization:** Mercy Jelagat Karoney, Mathew Kirtptonui Koech, Evangeline Wawira Njiru, Willis Dixon Owino Ong'or.

**Data curation:** Mercy Jelagat Karoney, Willis Dixon Owino Ong'or.

**Formal analysis:** Mercy Jelagat Karoney.

**Funding acquisition:** Mercy Jelagat Karoney.

**Investigation:** Mercy Jelagat Karoney, Mathew Kirtptonui Koech, Evangeline Wawira Njiru, Willis Dixon Owino Ong'or.

**Methodology:** Mercy Jelagat Karoney, Mathew Kirtptonui Koech, Evangeline Wawira Njiru.

**Project administration:** Mercy Jelagat Karoney, Mathew Kirtptonui Koech, Willis Dixon Owino Ong'or.

**Resources:** Mercy Jelagat Karoney.

**Software:** Mercy Jelagat Karoney.

**Supervision:** Evangeline Wawira Njiru, Willis Dixon Owino Ong'or.

**Validation:** Mercy Jelagat Karoney, Mathew Kirtptonui Koech, Evangeline Wawira Njiru, Willis Dixon Owino Ong'or.

**Writing – original draft:** Mercy Jelagat Karoney, Mathew Kirtptonui Koech, Evangeline Wawira Njiru, Willis Dixon Owino Ong'or.

**Writing – review & editing:** Mercy Jelagat Karoney, Mathew Kirtptonui Koech, Evangeline Wawira Njiru, Willis Dixon Owino Ong'or.

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
