## [Decision Letter · Decision Letter 0]

7 Jun 2022

PONE-D-21-36700Proximal tubular renal dysfunction among HIV infected patients on Tenofovir versus Tenofovir sparing regimen in western KenyaPLOS ONE

Dear Dr. Karoney,

Thank you for submitting your manuscript to PLOS ONE. After careful consideration, we feel that it has merit but does not fully meet PLOS ONE’s publication criteria as it currently stands. Therefore, we invite you to submit a revised version of the manuscript that addresses the points raised during the review process.

Please address all the issues raised by both the reviewers for further consideration of the mansucript.

We look forward to receiving your revised manuscript.

Kind regards,

Prasun K Datta, Ph.D

Academic Editor

PLOS ONE

Journal Requirements:

4. Please amend the manuscript submission data (via Edit Submission) to include author Evangeline Wawira Njiru.

Reviewers' comments:

Reviewer's Responses to Questions

**Comments to the Author**

1. Is the manuscript technically sound, and do the data support the conclusions?

Reviewer #1: Partly

Reviewer #2: Yes

2. Has the statistical analysis been performed appropriately and rigorously? 

Reviewer #1: No

Reviewer #2: Yes

3. Have the authors made all data underlying the findings in their manuscript fully available?

Reviewer #1: No

Reviewer #2: Yes

4. Is the manuscript presented in an intelligible fashion and written in standard English?

Reviewer #1: Yes

Reviewer #2: Yes

5. Review Comments to the Author

Reviewer #1: The authors aimed to assess proximal tubular, renal dysfunction, global renal function as markers of kidney functions among patients on TDF versus TDF-sparing regimen. I have some question and concerns below.

1. Was there any consideration for co-infections in the study?

2. Several calculation errors in table 1 need to be addressed. For example, should total be 516?

3. For the “occupation” row (table 1) it is unclear what the p value relates to.

4. Within the participant characteristics and clinical characteristics categories in Table 1 and 2, it would be useful to use symbols e.g, asterisk to indicate what specific comparisons the p values refer to.

5. With reference to the statement “Participants in the TDF-sparing group had more preexisting comorbidities compared to the TDF regimen group, 13.3% versus 6.1% p=0.02” is this an aggregate of a group of comorbidities? Or specifically to hypertension?

6. There are major flaws with the analysis approach. The authors tend to assign covariables categorically, which is not always appropriate. The primary focus and outcome could be “proximal tubular renal dysfunction” and other surrogates of renal dysfunction with TDF regimen as variable within and across groups (given the limitation in size for a modestly rare event). I recommend consultation with an advanced statistician. Associations between important variables should be backed up with p values in a univariate model, followed by multivariable analysis to identify independent variables associated with measures of renal dysfunction in HIV.

7. Discussion should be framed around factors that independently or not associated with makers of renal dysfunction, and the incidence of renal diseases in HIV-negative subjects within similar geography or ethnicity.

Reviewer #2: MJ Karoney et al. provide a sectional study article about significant change in PTRD among HIV infected patients treated with TDF against TDF spared patients. This article exhibits: 1) The proportion of PTRD is higher in the TDF treated patients compared to the TDF sparing group. 2) The profile of PTRD is measured in terms of normoglycemic glucosaria, metabolic acidosis, tubular proteinuria and phosphate wasting.

Comment: The manuscript is well written and describe every possible details of the study design clearly to propose the hypothesis. It is an important study to see the subclinical physiological effect of TDF as an important component of the ART regimen against HIV treatment. The study report is at par with the other possible reports related to TDF in response to the PTRD and the secondary manifestation due to it (GFR) irrespective of the socioeconomic demographic variability. However some minor points need to be reinforced to claim the publication.

Minor comments:

1) A brief note is needed to explain CKD-EPI formula even though the reference is cited in the manuscript.

2) In case of co morbidities hypertension, diabetes, kidney diseases were taken as the clinical characteristics in table 2. An explanation is needed whether any HIV related co-infections were also been taken into account as clinical characteristics to decide co-morbidities.

3) Usually PTRD during ART is at the subclinical level and reversible in nature. So, it is better that the authors may give an idea about the implication of this report in relation to the future prospect of the study.

6. PLOS authors have the option to publish the peer review history of their article (what does this mean?). If published, this will include your full peer review and any attached files.

Reviewer #1: No

Reviewer #2: No

---

## [Author Response · Author response to Decision Letter 0]

20 Jul 2022

Please see the response to each point raised by the academic editor and reviewer below. 

Reviewer 1:

1. Was there any consideration for co-infections in the study?

 No other co infections were considered in the study. The patients were ambulatory healthy participants mostly attending clinic for refills. The study did not budget for checking for co-infections and this may be considered a limitation 

2. Several calculation errors in table 1 need to be addressed. For example, should total be 516? This has been addressed and corrected

3. For the “occupation” row (table 1) it is unclear what the p value relates to. 

Please note that the authors have removed the demographic variables which we feel did bot add value to the tables presented or the overall paper

4. Within the participant characteristics and clinical characteristics categories in Table 1 and 2, it would be useful to use symbols e.g, asterisk to indicate what specific comparisons the p values refer to. – this has been done as suggested

5. With reference to the statement “Participants in the TDF-sparing group had more preexisting comorbidities compared to the TDF regimen group, 13.3% versus 6.1% p=0.02” is this an aggregate of a group of comorbidities? Or specifically to hypertension? The comorbidities here are diabetes and hypertension only this has been clarified in the table and text with results

6. There are major flaws with the analysis approach. The authors tend to assign covariables categorically, which is not always appropriate. Reanalysis was done with consultation of an advanced statistician The primary focus and outcome could be “proximal tubular renal dysfunction” and other surrogates of renal dysfunction with TDF regimen as variable within and across groups (given the limitation in size for a modestly rare event). I recommend consultation with an advanced statistician. Associations between important variables should be backed up with p values in a univariate model, followed by multivariable analysis to identify independent variables associated with measures of renal dysfunction in HIV. – the variables included in the final logistic regression were already decided a priori and therefore included in the final model despite some not having significant P values. The P values were not indicated for Univariate models because OR with 95% CI were used to show significance

7. Discussion should be framed around factors that independently or not associated with makers of renal dysfunction, and the incidence of renal diseases in HIV-negative subjects within similar geography or ethnicity. These factors were those that were a priori decided upon by the authors. The discussion highlights the subclinical nature of the PTRD with not much emphasis on the factors associated because they were not found to be significant in this study.

Reviewer #2: MJ Karoney et al. provide a sectional study article about significant change in PTRD among HIV infected patients treated with TDF against TDF spared patients. This article exhibits: 1) The proportion of PTRD is higher in the TDF treated patients compared to the TDF sparing group. 2) The profile of PTRD is measured in terms of normoglycemic glucosaria, metabolic acidosis, tubular proteinuria and phosphate wasting.

Comment: The manuscript is well written and describe every possible details of the study design clearly to propose the hypothesis. It is an important study to see the subclinical physiological effect of TDF as an important component of the ART regimen against HIV treatment. The study report is at par with the other possible reports related to TDF in response to the PTRD and the secondary manifestation due to it (GFR) irrespective of the socioeconomic demographic variability. However some minor points need to be reinforced to claim the publication.

Minor comments:

1) A brief note is needed to explain CKD-EPI formula even though the reference is cited in the manuscript. Added 

2) In case of co morbidities hypertension, diabetes, kidney diseases were taken as the clinical characteristics in table 2. An explanation is needed whether any HIV related co-infections were also been taken into account as clinical characteristics to decide co-morbidities. Co-infections were not taken into account

3) Usually PTRD during ART is at the subclinical level and reversible in nature. So, it is better that the authors may give an idea about the implication of this report in relation to the future prospect of the study. – this has been added in the conclusion section

---

## [Editor Report · Decision Letter 1]

4 Aug 2022

Proximal tubular renal dysfunction among HIV infected patients on Tenofovir versus Tenofovir sparing regimen in western Kenya

PONE-D-21-36700R1

Dear Dr. Karoney,

We’re pleased to inform you that your manuscript has been judged scientifically suitable for publication and will be formally accepted for publication once it meets all outstanding technical requirements.

Kind regards,

Prasun K Datta, Ph.D

Academic Editor

PLOS ONE
---

## [Editor Report · Acceptance letter]

26 Aug 2022

PONE-D-21-36700R1 

Proximal tubular renal dysfunction among HIV infected patients on Tenofovir versus Tenofovir sparing regimen in western Kenya 

Dear Dr. Karoney:

I'm pleased to inform you that your manuscript has been deemed suitable for publication in PLOS ONE. Congratulations! Your manuscript is now with our production department. 

Kind regards, 

on behalf of

Dr. Prasun K Datta 

Academic Editor

PLOS ONE